# Universal and efficient extraction of lithium for lithium-ion battery recycling using mechanochemistry

Oleksandr Dolotko [1,2✉], Niclas Gehrke[1], Triantafillia Malliaridou[1], Raphael Sieweck[1], Laura Herrmann[1,3], Bettina Hunzinger[1], Michael Knapp [1] & Helmut Ehrenberg [1,2]

The increasing lithium-ion battery production calls for profitable and ecologically benign technologies for their recycling. Unfortunately, all used recycling technologies are always associated with large energy consumption and utilization of corrosive reagents, which creates a risk to the environment. Herein we report a highly efficient mechanochemically induced acid-free process for recycling Li from cathode materials of different chemistries such as $LiCoO_2$, $LiMn_2O_4$, $Li(CoNiMn)O_2$, and $LiFePO_4$. The introduced technology uses Al as a reducing agent in the mechanochemical reaction. Two different processes have been developed to regenerate lithium and transform it into pure $Li_2CO_3$. The mechanisms of mechanochemical transformation, aqueous leaching, and lithium purification were investigated. The presented technology achieves a recovery rate for Li of up to 70% without applying any corrosive leachates or utilizing high temperatures. The key innovation is that the regeneration of lithium was successfully performed for all relevant cathode chemistries, including their mixture.

[1] Karlsruhe Institute of Technology (KIT), Institute for Applied Materials-Energy Storage Systems (IAM-ESS), Hermann-von-Helmholtz-Platz 1, D-76344 Eggenstein-Leopoldshafen, Karlsruhe, Germany. [2] Helmholtz-Institute Ulm for Electrochemical Energy Storage (HIU), P.O. Box 3640, D-76021 Karlsruhe, Germany. [3] EnBW Energie Baden-Württemberg AG, Durlacher Allee 93, 76131 Karlsruhe, Germany. ✉email: oleksandr.dolotko@kit.edu

Lithium-ion batteries (LIBs) have experienced a leap in their development, especially with shifting their application from small consumer electronics to the market of electric vehicles and energy storage power batteries[1]. The growth of the use and production imposes the need for infrastructure and strategies to handle LIB waste and potentially recover precious components of batteries without irreversible pollution and environmental damage. The recycling industry is currently unprepared to handle the large volumes of end-of-life batteries and production scrap that will need to be recycled in the near future. This capacity needs to be developed over the next few years. Today, the primary materials recycled are the cathode materials nickel and cobalt, the current collector materials copper and aluminum, and other passive components such as steel. Recycling of lithium is, however, currently expensive and, in many cases, not profitable[2–6]. Despite the intensive research activity and progress in the industrial sector, the recycling technology for LIBs remains in its infancy and requires significant development. Currently, most of the recycling technologies are based on pyrometallurgy, hydrometallurgy, or biohydrometallurgy processes.

The pyrometallurgical process transforms the spent LIBs into alloys containing d-elements and slag products (lithium-rich slag) at temperatures higher than 1000 °C[7,8]. Using different slag modification agents ($SiO_2$, $CaO$, $Al_2O_3$, etc.), the phase composition of the slag can be adjusted, while the alloy products are further recovered via subsequent hydrometallurgy treatment[9–11]. The main advantage of the pyrometallurgical process is the absence of a raw material pre-treatment step. However, it is always accompanied by significant investment into equipment, energy-wasting, and heavy pollution. Furthermore, although the pyrometallurgy process can selectively enrich lithium in the slag phase, the direct leaching of lithium from slag requires high energy consumption[12,13]. Many companies and academic researchers have developed hydrometallurgical processes in response to these problems. This technology has the advantage of low exhaust emission, mild reaction conditions, and high metal recovery efficiencies. With the goal of selective separation of the valuable metal ions in the solution and preparation of the corresponding raw materials, the typical hydrometallurgy process mainly includes three major process steps. In the first step, which comprises leaching, all metals are dissolved with the help of an acid, base, or salt. The following second step includes the purification of the metals using selective chemical reactions, such as precipitation, ion exchange, liquid-solid, liquid-liquid reaction, solvent extraction, etc. And in the last step, the targeted elements are recovered from solutions as a solid product via ionic precipitation, crystallization, or electrochemical reduction[14–17].

The complex leaching solution produced along the process often causes difficulties with the subsequent extraction and purification steps. One of the biggest challenges is the loss of metal ions due to co-extraction when removing or extracting target metal ions. The loss of lithium is one of the more typical examples. According to reports, over 20% of lithium ions are extracted simultaneously with nickel, cobalt, and manganese ions, and this part of lithium loss is challenging to recover further[18,19]. Despite the ability to produce high-quality products, hydrometallurgy is opposed by the complexity of the processes, which strongly depend on electrode chemistry and produces a significant amount of harmful waste.

Compared to pyrometallurgy which is always accompanied by emissions and energy consumption, and hydrometallurgy, with its complexity and waste generation, the biohydrometallurgical approach seems more favorable. This technology does not require adding toxic chemicals, thereby avoiding the generation of hazardous byproducts[20–23]. Bioleaching technology is only one-third of the cost of traditional leaching technology, more efficient and conducive to environmental protection and resource conservation. It is a "greener" and more environmentally friendly process. However, it is still in its initial stage of development and requires considerable follow-up research to improve process efficiency, scalability, and separability.

All present shortcomings of existing technologies are forcing the scientific community to find alternative methods for LIBs recycling. In response to all challenges, the mechanochemical (MC) approach in recycling processes receives more and more attention. The emerging MC technology induces chemical reactions between solid materials using mechanical forces such as grinding, extrusion, shearing, and friction[24]. This approach is successfully applied in recycling valuable materials from various electronic wastes due to its low cost, scalability, unique reaction mechanism, thermodynamics, and kinetic properties[25,26]. Furthermore, as chemical interactions in this process are activated by mechanical force and hazardous solvents are generally not employed, the MC approach is relatively safe and clean, with high reaction efficiency and low energy consumption[27,28].

Gradual recognition of its benefits in time, simplicity, cost, and less waste production broadens the MC application for LIBs recycling. In most cases, the MC step is utilized as pre-treatment to the battery materials, thus significantly improving the recovery of valuable components in the following hydrothermal process[29–32]. However, the most effective utilization of the MC approach is observed in processes when direct reactions between battery materials and additives occur[33–35]. Such technology enables the recovery of valuable metals at room temperature with a high extraction efficiency at ambient pressures and temperatures while avoiding corrosive solvents. Thus, in a recent publication, Dolotko et al. reported that solvent-free processing could successfully convert $LiCoO_2$ into metallic Co and Li-derivatives via reduction reactions mechanochemically[36]. Herein, using a similar approach, the systematic study of lithium recovery from the majority of the commercially used cathodes is presented. The aim of this work was the investigation of lithium recycling from $LiCoO_2$ (LCO), $Li(Ni_{0.33}Mn_{0.33}Co_{0.33})O_2$ (NMC), $LiMn_2O_4$ (LMO), $LiFePO_4$ (LFP), and their mixture by using the MC approach, where Al is used as a reducing agent for chemical transformation which is typically present as a current collector. The previously established lithium recycling process, which utilizes a mechanochemical reduction reaction, was further modified and improved in terms of its simplicity and possible industrial feasibility. Two different processes were developed and described. It was demonstrated that the proposed method could be called "universal," as it has a similar mechanism and can be applied for the majority of electrode chemistries while fostering excellent environmental sustainability and holding potential for reducing the overall costs of LIB recycling.

## Results and discussion
### Lithium extraction with process 1
*Process 1 for LCO cathode.* The recycling process 1, shown schematically in Fig. 1a, was applied for the $LiCoO_2$ material. XRD patterns in Fig. 2 show that ball milling of an equimolar mixture of $LiCoO_2$ and Al (1:1) for 3 h produces poorly crystalline material, in which a metallic Co-based composite (marked as metallic composite) with a cubic (fcc) structure can be vaguely distinguished by the presence of a broad reflection at ~45° 2θ. Due to its magnetic nature, metallic Co and its composites are easily extracted by a permanent magnet. Therefore, one of the visual signs of the reduction process of $LiCoO_2$ (which is non-magnetic) is the formation of the magnetic phase. Reduction of Co in $LiCoO_2$ by Al starts already after 30 min of milling, which is distinguishable by the appearance of the broad Bragg reflection at

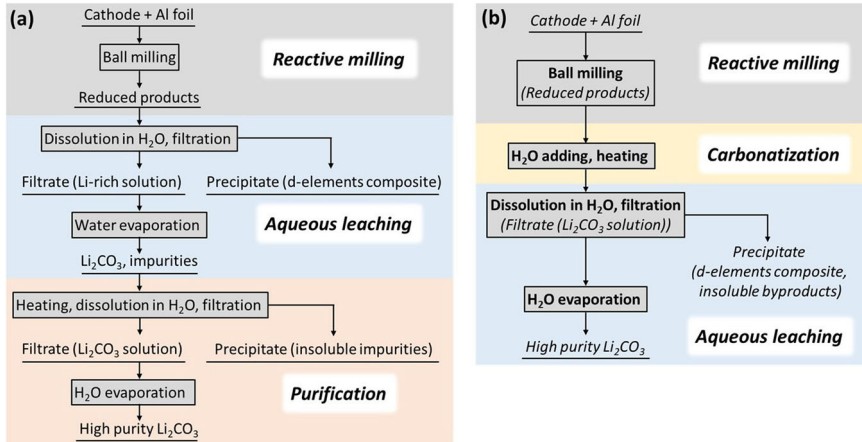

**Fig. 1 Flowsheets of the recycling processes for lithium extraction. a** process 1; **b** process 2.

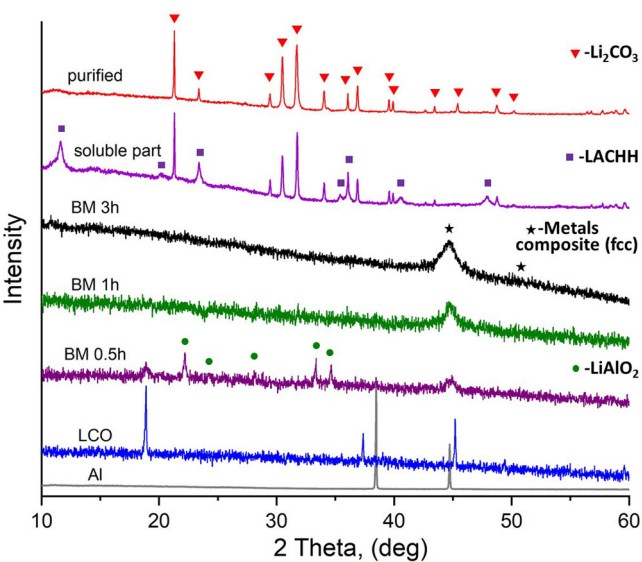

**Fig. 2 XRD patterns of the 1:1 molar mixture of LiCoO₂ and Al, measured after different ball milling times in a SPEX mill.** XRD patterns of starting materials LiCoO₂ and Al are presented for comparison. The most intensive Bragg reflections of intermediate and final products are marked for analysis.

~45° 2θ and the formation of the magnetic phase. Before the complete reduction, formation of the intermediate phase γ-LiAlO₂ (tetragonal structure, space group $P4_12_12$[37]) was observed after 30 min of MC reaction (Fig. 2). Continuation of the milling for 1 h and above, leads to the disappearance of the Bragg reflections of the γ-LiAlO₂ phase, while only the broad reflection of the metallic composite remains visible.

The MC reduction reaction of LiCoO₂ with Al as a reducing agent can be described by Eq. 1, where feasible reaction intermediates are given in parentheses:

$$2LiCoO_2 + 2Al \rightarrow Co + \{Li_2O + Al_2O_3\} \quad (1)$$

According to the XRD data, the newly formed oxides of lithium and aluminum oxides further interact with the formation of the γ-LiAlO₂ (Eq. 2)

$$Li_2O + Al_2O_3 \rightarrow 2LiAlO_2 \quad (2)$$

It remains unclear what happens to γ-LiAlO₂ upon prolonged milling, as all phases, except the metallic composite, become XRD amorphous.

The morphology change of the sample upon milling was investigated using the SEM method. The microscopy images indicate decreasing particle sizes and the development of a homogeneous mixture during the MC reaction (Figs. S1a–c).

The aqueous leaching process, followed by filtration of the solid residue, produces the soluble filtrate, which was recrystallized by water evaporation. According to the XRD analysis (Fig. 2, soluble part), the recrystallized product contains lithium carbonate (Li₂CO₃) and lithium aluminum carbonate hydroxide hydrate, Li₂Al₄(CO₃)(OH)₁₂·3H₂O (LACHH). The formation of these products can be explained by the reactions (Eqs. 3–5), which may take place simultaneously during aqueous leaching and drying in air atmosphere.

$$\{Li_2O + Al_2O_3\} + H_2O \rightarrow \{LiOH + Li_xAlO_x(OH)_z + Al_2O_3\} \quad (3)$$

$$2LiOH + CO_2 \rightarrow Li_2CO_3 + H_2O \quad (4)$$

$$\{Li_xAlO_x(OH)_z\} + H_2O + CO_2 \rightarrow Li_2Al_4(CO_3)(OH)_{12} \cdot 3H_2O \quad (5)$$

In the purification step, the lithium present in the recrystallized soluble part was transformed into lithium carbonate. It was achieved by heating the sample after aqueous leaching to 350 °C for 3 h in air atmosphere. According to literature data, the LACHH decomposing starts at 250–290 °C with the formation of Li₂CO₃ and Al₂O₃ (Eq. 6)[38]. Our research confirms this decomposition route at 350 °C:

$$Li_2Al_4(CO_3)(OH)_{12} \cdot 3H_2O \rightarrow Li_2CO_3 + 2Al_2O_3 + 9H_2O \uparrow \quad (6)$$

The XRD analysis of the intermediate products of the purification process shows that heating the soluble fraction after the aqueous leaching (Fig. S2, leached soluble part) to 350 °C for 3 h leads to the vanishing of the reflections of LACHH, where only Li₂CO₃ is distinguishable (Fig. S2, 350 °C).

The heat-treated sample was dispersed into water and filtrated in the subsequent step. In this step, water-soluble Li₂CO₃ was separated and purified from the water-insoluble Al₂O₃ (Fig. 2, purified and Fig. S2, 350 °C-soluble). The solid residue, which has poor crystallinity (Fig. S2, 350 °C-insoluble), was heat-treated at 700 °C for 12 h in air. Recrystallized γ-Al₂O₃ (space group Fd-3m) was determined after heating, which supports the decomposition of LACHH to Li₂CO₃ and Al₂O₃ via Eq. 6 (Fig. S2, 350 °C-insoluble-heated).

*Process 1 for NMC cathode.* A similar process was applied to recycle lithium from NMC material. The mechanism of reduction reaction of the $Li(Ni_{0.33}Mn_{0.33}Co_{0.33})O_2$ by Al is analogous to the $LiCoO_2$. In general, it can be expressed by Eq. 7, where M corresponds to the metallic composite, which contains all d-metals present in the starting cathode material:

$$2LiMO_2 + 2Al \rightarrow 2M + \{Li_2O + Al_2O_3\} \qquad (7)$$

Formation of the intermediate phase $LiAlO_2$ was also observed in this system after 30 min of reaction, which is still present after 1 h of milling. At the same time, starting materials NMC and Al coexist with intermediate and final products. After a prolonged MC treatment, they become not detectable by XRD. After 3 h of milling, only broad Bragg reflections of the metallic composite with a cubic (fcc) structure are distinguishable in the XRD pattern (Fig. S3). The SEM investigation shows decreased sizes and destroyed uniformity of the particles upon continuous milling (Fig. S4), leading to broadened reflections in the powder diffraction pattern.

The water-soluble part at room temperature contains $Li_2CO_3$ and LACHH phases after its recrystallization at 70 °C. The single-phase $Li_2CO_3$ was obtained after the purification process described above for the LCO-Al system (Fig. S3).

*Process 1 for LMO cathode.* Following the recycling process 1, lithium was extracted from the commercial $LiMn_2O_4$ material. In order to reduce all manganese ($Mn^{+3}$ and $Mn^{+4}$) present in the lithium manganese oxide to the $Mn^0$, the molar ratio of the components $LiMn_2O_4$ to Al was selected as 1: 2.33 (Eq. 8):

$$2LiMn_2O_4 + 4.66Al \rightarrow 4Mn + \{Li_2O + 2.33Al_2O_3\} \qquad (8)$$

The ball milling of the starting materials for 30 min leads to the formation of the $LiAlO_2$ compound. As the general amount of lithium in this system is lower than in the NMC- and LCO-containing mixtures, the intensity of the reflections of the $LiAlO_2$ phase is much lower but still distinguishable. The higher amount of aluminum in the mixture leads to increased reflection intensity of the $\gamma$-$Al_2O_3$ (space group Fd-3m) product after 1 and 3 h of milling (Fig. S5). The XRD pattern of the final product also contains broad Bragg reflections, which correspond to the cubic ccp structure, in which $Mn^0$ usually crystallizes. There is a possibility that some Al atoms were introduced into the structure or cold-welded. Therefore in Fig. S5, this phase is designated a metal composite. The SEM characterization of the mixture, milled for different times, shows that the particle size decreases already after 30 min. Further milling does not significantly change the sample microstructure (Fig. S6).

A difference compared to the other two systems described above was observed in the composition of the soluble fraction (Fig. S5). The presence of $Al(OH)_3$ (monoclinic structure, space group $P2_1/c$), which coexists with minor amounts of $Li_2CO_3$ and LACHH was detected by XRD analysis. The possible interaction of the LiOH with aluminum-containing products upon water leaching can explain the formation of the aluminum hydroxide. The following purification step leads to separating all Al-containing components from $Li_2CO_3$. As a result, pure lithium carbonate remains at the end of process 1 (Fig. S5).

*Process 1 for LFP cathode.* Process 1 for lithium recycling was also applied to the LFP cathode material. The molar ratio of starting materials $LiFePO_4$ and Al was selected as 1:3. The XRD phase analysis shows the beginning of the reduction reaction after 30 min of ball milling (Fig. 3). The formation of the $LiAlO_2$, $Al_2O_3$, and $Fe_2P$ (space group P-62m) compounds was identified. Further MC treatment increases the amount of the final products $Al_2O_3$ and $Fe_2P$, while the phase $LiAlO_2$ becomes undetectable

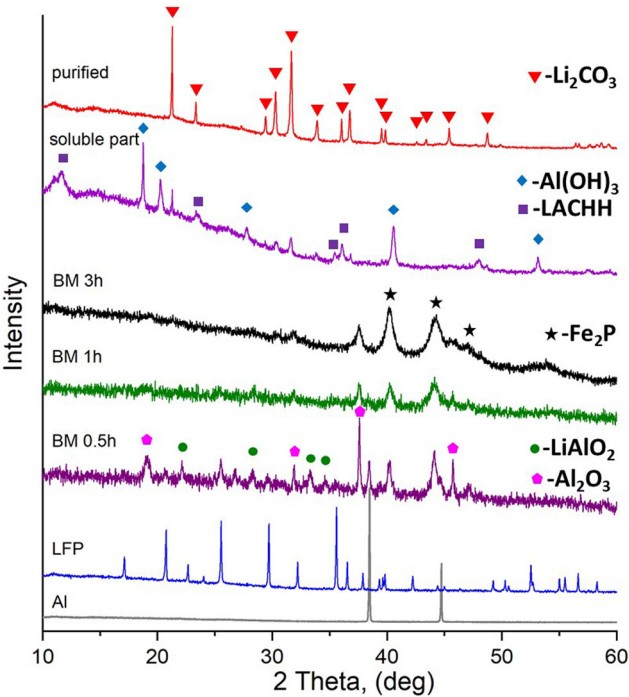

**Fig. 3 XRD patterns of the 1:3 molar mixture of $LiFePO_4$ and Al measured after different ball milling times in a SPEX mill.** XRD patterns of starting materials $LiFePO_4$ and Al are presented for comparison. The most intensive Bragg reflections of intermediate and final products are marked for analysis.

after 3 h of reaction. Based on the phase composition of the final product, the MC reduction reaction of $LiFePO_4$ performed in air atmosphere can be presented as:

$$2LiFePO_4 + 6Al + 0.25O_2 \rightarrow Li_2O + Fe_2P + AlP + 2.5Al_2O_3 \qquad (9)$$

One of the possible products of this reaction, AlP, is not detectable by XRD analysis, which can be explained by the poor crystallinity of the ball-milled materials. The SEM characterization of the milled products of the $LiFePO_4$-3Al system confirms this idea, showing significant microstructure changes and decreased particle size during prolonged MC reaction (Fig. S7).

The XRD analysis of the recrystallized water-soluble products of the aqueous leaching shows the presence of $Al(OH)_3$, LACHH, and $Li_2CO_3$ (Fig. 3, soluble part). However, only $Li_2CO_3$ was present in the recycled product after the purification step, when the soluble part was heated to 350 °C for 3 h, washed, and filtrated (Fig. 3, purified).

*Process 1 for a mixture of cathodes.* The versatility of the developed method was investigated by applying process 1 to recycle Li from a mixture of different cathodes. The mix of components, taken in the molar ratio of LCO:NMC:LMO:LFP:7.33Al, was ball-milled at different times for up to 3 h. The reduction reaction of the mixture is completed with the formation of the XRD detectable metal composite (fcc-structure), $Fe_2P$, and $Al_2O_3$ (Fig. 4). The intermediate $LiAlO_2$ phase forms after 0.5 h of milling and becomes undetectable in the following steps. The SEM characterization of the mixture, MC treated for different times, shows significant morphology changes upon milling (Fig. S8). Decreased particle sizes and homogenization of the mixture was obtained upon continuous milling. The water-soluble fraction after crystallization at 70 °C contains a mixture of $Li_2CO_3$ and $Al(OH)_3$ (Fig. 4, soluble part). No LACHH phase was detectable at this step. The purification process leads to removing

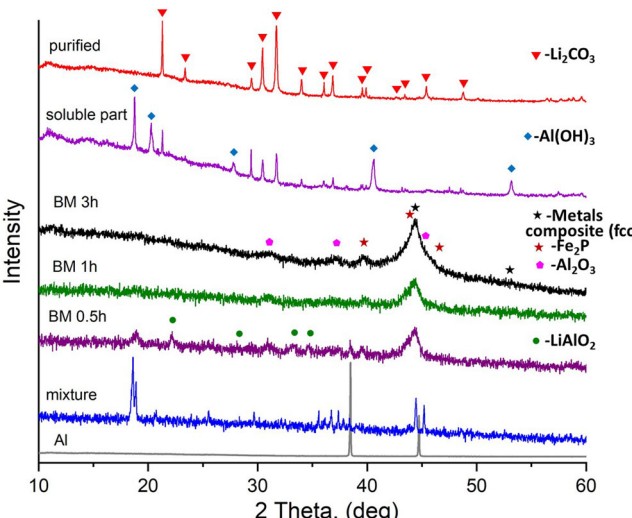

**Fig. 4 XRD patterns of the mixture of LiCoO₂ (LCO), Li(Ni₀.₃₃Mn₀.₃₃Co₀.₃₃)O₂ (NMC), LiMn₂O₄ (LMO), and LiFePO₄ (LFP) cathodes with Al measured after different ball milling times in a SPEX mill.** XRD patterns of starting materials - cathode physical mixture and Al, are presented for comparison. The most intensive Bragg reflections of intermediate and final products are marked for analysis.

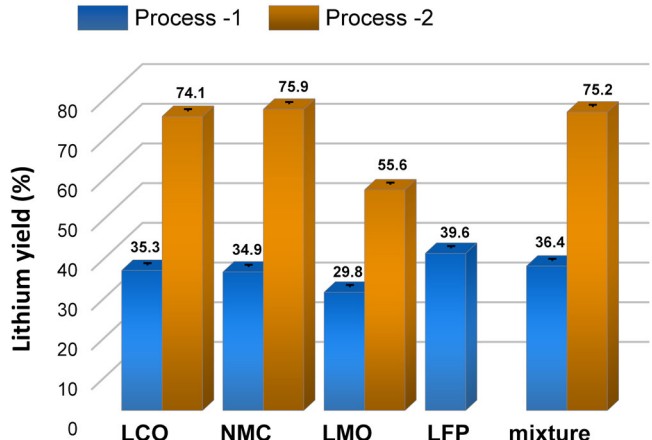

**Fig. 5 Comparison of the lithium recycling proficiency for different cathode materials.** LiCoO₂ (LCO), Li(Ni₀.₃₃Mn₀.₃₃Co₀.₃₃)O₂ (NMC), LiMn₂O₄ (LMO), LiFePO₄ (LFP), and their mixture at different processes.

Al-containing components and forming the pure $Li_2CO_3$ (Fig. 4, purified).

The calculated lithium recovery rate for process 1 was observed to be 30–40%, depending on the cathode material used (Fig. 5). The relatively low lithium yield can be related to the fact that not all lithium is transformed into water-soluble compounds after the ball milling. The reason might be the formation of an intermediate compound, $LiAlO_2$, formed after 30 min of milling that can amorphize upon prolonged MC treatment. As a result, it can become XRD-amorphous, therefore, not detectable. This compound might only be partially soluble in water at room temperature and thus remains in the solid residue after water leaching.

**Lithium extraction with process 2**. In order to recover all available lithium in the mixture after the reduction reaction, the "Carbonatization" step was introduced into a recycling flowsheet, as shown in Fig. 1b (process 2). The idea behind this step is to

transform $LiAlO_2$, which might be present in the mixture in an amorphous state after the MC reaction, into the LACHH compound via the interaction of $LiAlO_2$ with added water and $CO_2$ in ambient air in the presence of other products by reaction 10:

$$4LiAlO_2 + 9H_2O + 2CO_2 \rightarrow Li_2Al_4(CO_3)(OH)_{12} \cdot 3H_2O + Li_2CO_3 \quad (10)$$

Within the "Carbonatization" step, the mixture containing reduction products, $Li_2CO_3$ and LACHH is heated to decompose LACHH to $Al_2O_3$ and $Li_2CO_3$ (Eq. 6). In the following room temperature aqueous leaching step, the only water-soluble constituent $Li_2CO_3$ is separated from other products by filtration and recrystallization at 70 °C.

*Process 2 for LCO, NMC, and LMO cathodes*. The XRD analysis shows a similar mechanism of reactions in process 2 for all three types of cathodes (LCO, NMC, and LMO). The carbonatization step leads to the formation of LACHH phase, which coexists with the metal composites (Figs. S9–S11). At the same time, heating the metallic composite in air and the presence of water might lead to its partial surface oxidation. This is particularly pronounced in the LiMn₂O₄-2.33Al system, where the appearance of the $MnO_2$ and metallic composite with bcc structure was observed (Fig. S11, carbonatized).

In the following aqueous leaching step, pure lithium carbonate was obtained in all three systems (Figs. S9–S11, leached). A significant increase in Li yield was obtained in all three systems when process 2 was used (Fig. 5). The remarkable improvement in the Li recovery confirms the assumption that a fraction of lithium remains insoluble during the leaching procedure in process 1. Therefore, the carbonatization step is crucial for transforming almost all lithium available in the mixture into a carbonate salt and its extraction from the solid fraction.

*Process 2 for LFP cathode*. Different reaction products were obtained in the carbonization step in the LFP-3Al system. It was investigated by XRD analysis that heating the mixture after MC reduction in the presence of water, and air leads to the formation of $Al(OH)_3$ and a minor amount of LACHH (Fig. S12, carbonatized). As the amount of lithium in this system is low (theoretically ~3 wt%), the intensity of LACHH reflections is low and vaguely distinguishable. At the same time, a significant amount of aluminum in the basic solution leads to the formation of aluminum hydroxide, which is detected by XRD analysis. Two other reduction products, $Fe_2P$ and $Al_2O_3$, are still present after the carbonatization process (Fig. S12).

Phase analysis of the recrystallized products of the aqueous leaching shows the formation of $Li_2CO_3$, $Li_3PO_4$, $Fe_2(HPO_3)_3$, and $AlFe(PO_4)O$ (Fig. S12, leached). Such a complex multiphase composition of the mixture makes it difficult to distinguish the Li yield in process 2. Therefore, this value is not presented in Fig. 5. To separate lithium from other components, an additional purification step must be introduced. Such research is currently underway and will be published in a subsequent research article, where the LFP-Al system will be investigated in more detail.

*Process 2 for a mixture of cathodes*. A similar procedure of Li recycling using process 2 was applied to a mixture of different cathode materials, which were taken in the molar ratio with Al as LCO:NMC:LMO:LFP:Al as 1:1:1:1:7.33. The XRD patterns collected after different steps of process 2 are shown in Fig. S13. The main product of the carbonatization step is the LACHH compound, which coexists with the metallic composite and $Fe_2P$ (Fig. S13, carbonatized). Despite the presence of the $Fe_2P$ compound in the mixture before its carbonatization and aqueous leaching, the presence of $Li_3PO_4$ or other phosphate or phosphite salts formed in the LFP-Al system was not detected. The only

**Table 1 Chemical analysis of the obtained lithium carbonate, performed with inductively coupled plasma atomic emission spectroscopy (ICP-OES).**

|  | Impurities [µg/mg ($Li_2CO_3$)] | | | | | | Purity [wt%] |
|---|---|---|---|---|---|---|---|
|  | Al | Co | Ni | Mn | Fe | P | $Li_2CO_3$ |
| LCO-Pr1 | 3.08 | - | - | - | - | - | 99.69% |
| LCO-Pr2 | 1.31 | - | - | - | - | - | 99.87% |
| NMC-Pr1 | 0.82 | - | - | - | - | - | 99.92% |
| NMC-Pr2 | 1.24 | - | - | - | - | - | 99.88% |
| LMO-Pr1 | 0.74 | - | - | - | - | - | 99.93% |
| LMO-Pr2 | 4.35 | - | - | - | - | - | 96.40% |
| LFP-Pr1 | 0.97 | - | - | - | - | 73.3 | 93.10% |
| LFP-Pr2 | 4.28 | - | - | - | - | 388 | 71.81% |
| Mix-Pr1 | 0.38 | - | - | - | - | 8.11 | 99.16% |
| Mix-Pr2 | 0.62 | - | - | - | - | 2.15 | 99.72% |

Calculated purities and impurities of $Li_2CO_3$ derived from $LiCoO_2$-Al system, treated by Processes 1 ad 2 (LCO-Pr1; LCO-Pr2), NMC-Al system, treated by Processes 1 ad 2 (NMC-Pr1; NMC-Pr2), $LiMn_2O_4$-Al system, treated by Processes 1 ad 2 (LMO-Pr1; LMO-Pr2), $LiFePO_4$-Al system, treated by Processes 1 ad 2 (LFP-Pr1; LFP-Pr2), the mixture of all electrodes-Al system, treated by Processes 1 ad 2 (Mix-Pr1; Mix-Pr2) (Values that fall below the detection limit are not reported).

product obtained in the aqueous leaching and filtration step was pure $Li_2CO_3$ (Fig. S13, leached). It can be assumed that the formation of the soluble phosphorus salts depends on the solution's acidity and the amount of Al used in the reaction.

The calculated Li yield in this system with process 2 increased more than twice compared to process 1 (Fig. 5).

In addition to the significant increase in the Li recovery yield, process 2 decreases the number of steps in the recycling flowchart (Fig. 1), thus improving the economic viability of this recycling technique, where every step adds extra cost to the process.

The chemical analysis of the obtained lithium carbonate, performed by the ICP-OES method (Table 1), showed $Li_2CO_3$ of purity above 99 wt% can be obtained from LCO, NMC LMO, and the mixture of electrodes. The lowest purity was obtained for the LFP-Al system, with Al and P as the main impurity elements. The presence of phosphorus impurity in all LFP-containing systems can be explained by forming the $Li_3PO_4$ compound in the final product. The Al-containing impurity, which is not visible on the XRD powder diffraction patterns, can be related to the existence of the X-ray amorphous $Al_2O_3$, which partly penetrated through the filter paper. Thus, for improvement of the $Li_2CO_3$ purity, another water dissolution and filtration step might be necessary to introduce.

The further separation of the d-elements, obtained in a metallic composite state, will be subject to an upcoming publication.

A few essential issues must be considered if the presented technology will be selected for industrial use. It is known that battery materials can be supplied for recycling in different conditions. One possible candidate for recycling is "off-specification" powders of cathodes, which are not used in battery production due to their failed compositions or other parameters, which lead to sorting them out from the electrode production lines. The discovered technology presented in this article can be applied to these materials without significant adjustments. The reaction conditions and final recycling products are expected to be similar to the ones investigated in this work.

Another type of battery material for recycling can be the electrode scrap or black masses, which might contain other components in addition to the active cathode materials. These extra components, like a binder, graphitic anode, copper, or other

additives or side products of the black mass preparation, might affect the mechanochemically-induced recycling process. As theoretical and experimental investigations show, at thermodynamic-controlled interfaces, increasing contact areas and a number of interaction events positively influence the activity of reactive materials and are crucial to guarantee high-reaction kinetics[39–41]. Ball-milling is the process where powder particles are treated by repeated deformation, fracture, and welding by highly energetic collisions of grinding media. As a result, the surface area and interface area increased upon milling when a number of events of kinetic energy transfer from the milling tools into the reactive materials also increased. Changing the conditions of mechanochemical processes, such as ball-to-sample ratio, the geometry of the ball milling equipment, the presence of milling assistant agents, selection of reducer/oxidizer, and ball milling time, play a crucial role in designing the recycling process[42,43]. One of the possible adjustments to the proposed technology can be the increased ball milling time due to the presence of multiple non-reactive components in the reaction mixture. The components, 'inert' to the reaction, can play the role of 'insulators' for the materials taking part in the reduction process, thus may affecting ball milling conditions. An additional adjustment might be needed in the process of the purification of the final $Li_2CO_3$. The presence of the F-containing binder and electrolyte salts, like $LiPF_6$, might lead to the formation of LiF as an impurity in the recycled lithium carbonate. All these possibilities and challenges are currently under investigation, therefore, will be addressed in our future publications.

To summarize the advantages of the developed techniques, it is essential to emphasize the universality of their application. This is particularly important for its implementation in industry, where various suppliers provide waste batteries with different and often unknown chemistries. Furthermore, by introducing the investigated technique, the battery sorting procedure can be eliminated, making all processes applicable to a large variety of batteries.

## Conclusions

The mechanochemically induced Li recycling method from various primarily used cathode chemistries such as $LiCoO_2$, $LiMn_2O_4$, $Li(CoNiMn)O_2$, $LiFePO_4$, and their mixture was developed. Aluminum, the material of the current collector of the cathode, was used as a reducing agent, thus eliminating additional external additives in the recycling process. Two recycling methods were proposed and investigated. Both approaches start with a mechanochemical treatment of the cathode with Al, which leads to reduction reactions with the formation of metallic composites containing d-elements, aluminum oxide, and water-soluble lithium products.

The step of the aqueous leaching in process 1 leads to the formation of $Li_2CO_3$ in the mixture with LACHH. In the LFP-Al system, $Al(OH)_3$ was detected in this step. The following purification step, which includes heating, water solution, and filtration, leads to the decomposition of LACHH and $Al(OH)_3$, thus obtaining the pure $Li_2CO_3$.

The low Li yield in process 1 (29.8–39.6%) was explained by a loss of Li in the form of an insoluble component. Therefore, a carbonatization step was introduced in process 2, which reduced the number of steps and a significant increase in Li yield (55.6–75.9%).

The developed process is simple and energy efficient, thus, offering clear advantages over other known LIB recycling techniques. Furthermore, the method can be declared universal, as it can be applied to all currently used cathode chemistries separately as well as their mixtures. Thus, while utilized, this technique can avoid the sorting step in the recycling plant.

## Methods

**Materials**. $LiCoO_2$ (97 wt%) was purchased from Alfa Aesar, $LiMn_2O_4$ (>99%)—from Sigma-Aldrich, $LiFePO_4$ (~98.5%, coated with carbon)—from MTI Corporation and $Li(Ni_{0.33}Mn_{0.33}Co_{0.33})O_2$—from BASF, Germany. Aluminum foil (from Novelis) served as a source of Al in selected experiments. All materials were used as received.

**Lithium extraction procedure**. This study applied two lithium extraction processes schematically presented in Fig. 1. Both methods start with the *reactive milling* step, followed by *aqueous leaching* at room temperature for process 1. In the next step of this process, *purification* of the final product is performed. Process 2 after the reactive milling includes a *carbonatization* and heating procedure. This process is finished by *aqueous leaching* and obtains pure lithium carbonate ($Li_2CO_3$).

*Reactive milling*. About 2 g of a mixture of starting materials is prepared in an appropriate molar ratio and ball-milled for 0.5–3 h in a 65 ml hardened-steel vial with 20 g of steel balls using SPEX 8000 shaker mill. Al foil was cut into pieces of 1–2 cm size before the milling. All experiments were performed in air atmosphere.

*Aqueous leaching*. The milled (process 1) or carbonized (process 2) samples are mixed with deionized (DI) water and stirred for a few minutes in air. Then, the insoluble fraction of the obtained mixture was filtered through a paper filter using a vacuum pump. The filtrate was further heated on a heating plate to evaporate the major part of the water and dried entirely at 70 °C overnight.

*Purification*. To purify the $Li_2CO_3$ after the aqueous leaching in process 1, the obtained product was heated in the muffle furnace to 350 °C for 3 h in air. The heated sample was mixed with DI water and filtrated. The water-soluble part contains pure $Li_2CO_3$, which was further recrystallized by water evaporation.

*Carbonatization*. To transform the lithium compounds into carbonate in process 2, the ball-milled samples were mixed with DI water and heated to 90 °C for 1 h upon stirring. After that, the mixtures were dried at 70 °C overnight in air atmosphere.

**Materials characterization**. Phase analysis of the reaction products was carried out using X-ray powder diffraction (XRD) on a STOE Stadi P powder diffractometer with monochromatic Cu-Kα₁ radiation ($\lambda = 1.54056$ Å) in transmission geometry. The XRD measurements were performed at room temperature with a 0.015° 2θ step between 10 and 70 degrees of 2θ. The samples were prepared on a Kapton foil and the Kapton film's presence visibly adds amorphous-like background to the XRD patterns at $10° < 2\theta < 17°$. The microstructural properties of the materials were studied by using scanning electron microscopy (SEM). Images of the starting materials and the products after milling were collected using a MERLIN Scanning Electron Microscope from Carl Zeiss. The purity of obtained $Li_2CO_3$ and concentrations of Li, Al, Co, Ni, Mn, Fe, and P in the product were analyzed using inductively coupled plasma optical emission spectroscopy (ICP-OES, 700 Series: Agilent Technologies). The details of purity calculations are presented in the Supplementary Methods section of the Supporting Information (SI) file.

**Lithium yield calculation**. For the lithium recovery rate calculation, the ball-milled mixture with the known composition of starting materials was weighed and used throughout the whole recycling protocol without taking the sample in the intermediate steps for analysis. As no visible gas release was observed during the ball-milling process, the elemental composition of the ball-milled samples was considered unchanged. The weight of the finally obtained $Li_2CO_3$ was used for the yield calculation based on the theoretical amount of lithium present in the starting mixture.

## Data availability

The data supporting the findings of this study are available within this article and its Supplementary Information. Extra data are available from the corresponding author upon reasonable request.

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

## Acknowledgements

This work contributes to the research performed at CELEST (Center for Electrochemical Energy Storage Ulm-Karlsruhe), Germany.

## Author contributions

O.D. conceived the idea, supervised and guided the study, contributed to the design and performance of experiments, data interpretation, and manuscript writing; N.G. and T.M. performed experiments; R.S. and L.H. performed experiments for chemical analysis by ICP-OES; B.H. performed SEM measurements; M.K. and H.E. contributed to data interpretation and the manuscript writing. All authors provided comments and edits during the preparation of the manuscript.

## Funding

## Competing interests

The authors declare no competing interests.
