## [Peer Review File · Communications Chemistry]

Reviewers' comments:

Reviewer #1 (Remarks to the Author):

The article proposes a mechanochemical (MC) process to recover lithium from the black mass of end-of-life lithium-ion batteries. The process includes a MC reduction with metallic aluminum, which works as reducing compound, followed by the separation/recovery of lithium. To this aim, two different alternative recovery routes are proposed and tested to be implemented after the MC reaction stage: a first route including water leaching, filtration, and heating of filtered solid to 350 °C, and a second route proceeding through carbonatation. The evolution of transformations taking place over the successive process stages are analyzed by the extensive recourse to XRD phase analysis. This way, the different phases formed, and the mechanisms of reactions are identified.

I read the article with interest. The idea is fairly well written and structured and includes results that are both technological and scientific value. While the idea to perform a mechanochemical reduction with Al was already presented by the authors in a different paper (<https://doi.org/10.1016/j.jallcom.2020.153876>), here the application of the mechanochemical process is extended to all the main types of lithium-ion batteries. According to the results presented by the authors, the process can be considered general, as it can be applied to treat different types of lithium-ion batteries or a mixture of all the different types of batteries. An important element of novelty is the demonstration of an effective strategy for the recovery and purification of lithium, which permits separating the formed aluminum products.

However, before publication there are some issues that I recommend to address:

- My only concern is about the extensive use of XRD analysis to characterize the forming phases and products. XRD is indeed a valuable methodology to study the transformation mechanisms governing the process, but it should be coupled with a more detailed chemical characterization (including digestion and chemical analysis) aimed at quantifying the content of any element. While it may be accepted the absence of such an analysis for the intermediate products, it should be surely performed to assess the purity of the lithium carbonate. This is essential to evaluate the actual feasibility of the proposed process.
- How was the lithium recovery rate measured? The materials and methods section should be further extended by describing the details of the adopted procedure.
- There is an important issue that is not addressed by the present study: the presence of impurities in the treated samples. The authors use pure commercial cathodic materials to perform the experiments, while, in industrial practice, these materials invariably come to the process along with impurities (. While I understand that it is not the subject of the present study, I believe that it would be worth to mention the possible impact (whether they can have or not) of these impurities on process performances, and particularly on the purity of the final products. This should be, at least, considered an important issue to be addressed by future studies.
- I found Fig.1 illustrating the flowsheets of the proposed process routes not sufficiently clear. I would recommend, as it is common practice, to report a process flowsheet where the performed process operations are identified by blocks, possibly including an indication of the imposed operating conditions, and the generated streams are identified by arrows connecting the blocks. This can help the reader to rapidly figure out the sequence of operations every time process application with a new cathodic material is discussed.
- I would recommend to further clarify in the introduction section what was already done by the authors in previous studies and what are the main novelties of the present study. Not only the extension to different cathodic materials, but the efforts devoted to lithium recovery and purification are worth to be mentioned.

Therefore, publication of the article is recommended providing that the reported recommendations and comments are addressed.

Reviewer #2 (Remarks to the Author):

The authors report an efficient mechanochemically induced acid-free process for recycling Li from cathode materials. The proposed method is universal and interesting. However, it still can not meet the high standard of Communications Chemistry. Here are some comments:

1. The related method by mechanochemical and Al reduction has already been reported, such as doi.org/10.1016/j.jallcom.2020.153876. There is no scientific discovery on innovative chemical reaction or new mechanism.
2. The paper is more like an experimental report other than a scientific paper. Too much redundant information, e.g. there are 11 xrd patterns as figures in the MS. The writing should be improved.
3. Energy consumption for ball milling should be considered, although it is a universal method. The energy exchange efficiency for ball milling is very low.

Responses to reviewers:

Reviewer #1:

The article proposes a mechanochemical (MC) process to recover lithium from the black mass of end-of-life lithium-ion batteries. The process includes a MC reduction with metallic aluminum, which works as reducing compound, followed by the separation/recovery of lithium. To this aim, two different alternative recovery routes are proposed and tested to be implemented after the MC reaction stage: a first route including water leaching, filtration, and heating of filtered solid to 350 °C, and a second route proceeding through carbonatation. The evolution of transformations taking place over the successive process stages are analyzed by the extensive recourse to XRD phase analysis. This way, the different phases formed, and the mechanisms of reactions are identified.

I read the article with interest. The idea is fairly well written and structured and includes results that are both technological and scientific value. While the idea to perform a mechanochemical reduction with Al was already presented by the authors in a different paper (<https://doi.org/10.1016/j.jallcom.2020.153876>), here the application of the mechanochemical process is extended to all the main types of lithium-ion batteries. According to the results presented by the authors, the process can be considered general, as it can be applied to treat different types of lithium-ion batteries or a mixture of all the different types of batteries. An important element of novelty is the demonstration of an effective strategy for the recovery and purification of lithium, which permits separating the formed aluminum products.

However, before publication there are some issues that I recommend to address:

Comment: - *My only concern is about the extensive use of XRD analysis to characterize the forming phases and products. XRD is indeed a valuable methodology to study the transformation mechanisms governing the process, but it should be coupled with a more detailed chemical characterization (including digestion and chemical analysis) aimed at quantifying the content of any element. While it may be accepted the absence of such an analysis for the intermediate products, it should be surely performed to assess the purity of the lithium carbonate. This is essential to evaluate the actual feasibility of the proposed process.*

Response:

The additional ICP-OES analysis of Li_2CO_3 material, obtained after recycling processes, was performed and included in the revised manuscript.

Comment: - *How was the lithium recovery rate measured? The materials and methods section should be further extended by describing the details of the adopted procedure.*

Response:

The description of the lithium recovery rate calculation is added to the materials and methods section:

2.3 Lithium yield calculation

For the lithium recovery rate calculation, the ball-milled mixture with the known composition of starting materials was weighed and used throughout the whole recycling protocol without taking the sample in the intermediate steps for analysis. As no visible gas release was observed during the ball-milling process, the elemental composition of the ball-milled samples was considered unchanged. The weight of the finally obtained Li_2CO_3 was used for the yield calculation based on the theoretical amount of lithium present in the starting mixture.

Comment:- *There is an important issue that is not addressed by the present study: the presence of impurities in the treated samples. The authors use pure commercial cathodic materials to perform the experiments, while, in industrial practice, these materials invariably come to the process along with impurities (. While I understand that it is not the subject of the present study, I believe that it would be worth to mention the possible impact (whether they can have or not) of these impurities on process performances, and particularly on the purity of the final products. This should be, at least, considered an important issue to be addressed by future studies.*

Response:

The reviewer's comment is valuable and raises the critical question of implementing the discovered technology at the industrial level. This point is discussed in the manuscript:

A few essential issues must be addressed and considered if the presented technology is selected for industrial use. It is known that battery materials can be supplied for recycling in different conditions. One possible candidate for recycling is "off-specification" powders of cathodes, which are not used in battery production due to their failed compositions or other parameters, which lead to sorting them out from the electrode production lines. The newly discovered technology presented in this article can be applied to these materials without significant adjustments. The reaction conditions and final recycling products are expected to be similar to the investigated in this work.

Another type of battery material for recycling can be the electrode scrap or black masses, which might contain other components in addition to the active cathode materials. These extra components, like a binder, graphitic anode, copper, or other additives or side products of the black mass preparation, might affect the mechanochemically-induced recycling process. One of the possible adjustments to the proposed technology can be the increased ball milling time due to the presence of multiple non-reactive components in the reaction mixture. The present "inert" for the reduction reaction components play the role of "insulators" for the materials, which take part in the reduction reaction, thus affecting ball milling conditions. An additional adjustment might be needed in the process of the purification of the final Li_2CO_3 . The presence of the F-containing binder and electrolyte salts, like LiPF_6 , might lead to the formation of LiF as an impurity in the recycled lithium carbonate. All these possibilities and challenges are currently under investigation, therefore, will be addressed in our future publications.

Comment:- *I found Fig.1 illustrating the flowsheets of the proposed process routes not sufficiently clear. I would recommend, as it is common practice, to report a process flowsheet where the performed process operations are identified by blocks, possibly including an indication of the imposed operating conditions, and the generated streams are identified by arrows connecting the blocks. This can help the reader to rapidly figure out the sequence of operations every time process application with a new cathodic material is discussed.*

Response:

Fig. 1 is modified as per the reviewer's comment.

Comment:- *I would recommend to further clarify in the introduction section what was already done by the authors in previous studies and what are the main novelties of the present study. Not only the extension to different cathodic materials, but the efforts devoted to lithium recovery and purification are worth to be mentioned.*

Response:

The introduction section is modified according to the reviewer's suggestions:

The previously established lithium recycling process, which utilizes a mechanochemical reduction reaction, was further modified and improved in terms of its simplicity and possible industrial feasibility. Two different processes were developed and described.

Therefore, publication of the article is recommended providing that the reported recommendations and comments are addressed.

Reviewer #2:

The authors report an efficient mechanochemically induced acid-free process for recycling Li from cathode materials. The proposed method is universal and interesting. However, it still can not meet the high standard of Communications Chemistry. Here are some comments:

Comment:- *The related method by mechanochemical and Al reduction has already been reported, such as doi.org/10.1016/j.jallcom.2020.153876. There is no scientific discovery on innovative chemical reaction or new mechanism.*

Response:

We respectfully disagree with the reviewer's comment. In addition to the usage of LCO, NMC, and LMO cathode chemistries, which have a similar mechanism of transformation, this work, it is presented the recycling of the LFP cathode, which has a different chemistry and reaction mechanism.

Another novelty of the current research is the discovery of a new, highly efficient, simple way of purification of lithium after water leaching (Process 2). These critical points are addressed in the introduction and discussion sections of the revised manuscript.

Comment:- *The paper is more like a experimental report other than a scientific paper. Too much redundant information, e.g. there are 11 xrd patterns as figures in the MS. The writing should be improved.*

Response: As suggested by the reviewer, the manuscript was revised, and changes along with additional discussion sections were added. All changes within the text are highlighted with red color.

3. Energy consumption for ball milling should be considered, although it is an universal method. The energy exchange efficiency for ball milling is very low.

Response:

We agree with the reviewer that the energy consumption of the ball milling process should be considered. This factor has to be considered in the stage of the process feasibility analysis for industrial usage, which is not a topic of the current research. However, to favor of the presented method, mechanochemical treatment is used in the powdering steps of almost all recycling processes. Therefore, the energy consumption for mechanochemical treatment is already partially included in the currently used industrial processes.

REVIEWERS' COMMENTS:

Reviewer #1 (Remarks to the Author):

The revisions introduced by the authors have adequately addressed the issues raised by the previous review. Publication of the article in the present form is therefore recommended.